# Leveraging Knowledge Distillation to Mitigate Model Collapse

## Abstract

Since the amount of data generated by neural networks on the Internet is growing rapidly due to widespread access to corresponding models, it is logical to inquire about the impact of this surge in synthetic data on the training of subsequent models that will utilize it during training. Previous work has demonstrated a concerning trend: models trained predominantly on synthetic data often experience a decline in performance, which can escalate to a complete loss of the ability to reproduce the initial distribution of real-world data. This phenomenon, now referred to as model collapse, highlights the potential pitfalls of over-reliance on synthetic datasets, which may lack the diversity and complexity inherent in genuine data. To address this issue, we propose a novel method that leverages the well-established technique of knowledge distillation. Our approach aims to mitigate the adverse effects of synthetic data by facilitating a more effective transfer of knowledge from high-performing teacher models to student model. By doing so, we seek to enhance not only the qualitative aspects—such as the richness and variability of the generated outputs—but also the quantitative metrics that gauge model performance. Through extensive experimentation, we demonstrate that our method improves the robustness and generalization capabilities of models trained on synthetic data, for instance for DDPM enhancement is 68.8%, in terms of the FID metric, contributing to a more sustainable and effective use of synthetic datasets in machine learning applications.

## 1 Introduction

Generative models have become ubiquitous, what has caused increase of synthetic data available. Consequently, future models will inevitably be trained on such kind of data, what can have detrimental effects on their performance — a phenomenon now commonly referred to as Model Collapse (Shumailov et al., 2023). As the proportion of synthetic data within a training subset increases, the behavior of the trained models becomes more unpredictable. For example, language models trained on synthetic data are prone to generate less likely or coherent responses compared to those trained on real-world data. Furthermore, it is not uncommon to observe repetitive words or phrases in the outputs of language models exposed to synthetic data during training.

The primary cause of this issue lies in the distributional properties of the synthetic data. Since synthetic data is generated from a finite number of samples, its distribution only approximates that of real data. As synthetic pipelines becomes deeper — through repeated cycles of data generation and model retraining — the resultant data distribution diverges progressively from the original, leading to degraded model performance over time.

Knowledge distillation (Hinton et al., 2015) has emerged as an effective technique for transferring the knowledge and capabilities of a larger model to a smaller one. This approach has shown considerable potential. (Sanh et al., 2020; Muralidharan et al., 2024). Motivated by this, we explore the application of knowledge distillation to mitigate the problem of Model Collapse by transferring the knowledge of a model trained on real data to a model that has been trained on synthetic data. In particular, since the models used in knowledge distilla-

tion in our case have the same number of parameters and architecture, we apply so-called self-distillation (Mobahi et al., 2020).

To sum up, the main contributions of this paper are:

1. We propose the use of knowledge distillation as a solution to address Model Collapse.
2. We perform experiments across different modalities and architectures, providing empirical evidence of the effectiveness of our approach.

Our research is structured as follows: at first we consider papers, related to ours, then describe the technique itself. After that we demonstrate the effectiveness of our method on unconditioned image generation task on such architectures as Variational Autoencoder (VAE) (Kingma & Welling, 2013) and Denoising diffusion probabilistic model (DDPM) (Ho et al., 2020). Further we consider causal language modelling task, i.e. abstractive text summarization. In conclusion, we discuss the obtained results.

## 2 Related work

In this section, we describe existing works, related to our.

**Model Collapse** (Shumailov et al., 2023) is a process of losing performance, while being progressively trained on synthetic data. The higher the percentage of synthetic data in the training dataset, the higher the divergence between the output distribution and the initial one. It has been showed, that this behavior is intrinsic for every modality and architecture, regardless of number of parameters, except that small models are more susceptible to collapse.

**Knowledge distillation** (Hinton et al., 2015) is an effective technique of transferring the capabilities of one trained model (teacher) to another with fewer parameters (student). It is achieved by adding to the initial loss function an extra term, that penalizes student for the difference between its outputs and teacher ones. In our case, architectures and numbers of parameters stay constant, and we use so-called self-distillation (Mobahi et al., 2020). But main difference of our approach is that the dataset used for training the student model consists completely of generated data.

**Approaches**, **aimed to mitigate model collapse** include several techniques. First of all, we can simply keep some percentage of the real-world data and do not replace it with synthetic, but accumulate all the data in one dataset (Gerstgrasser et al., 2024). This approach helps to reduce effect of Model Collapse, but it is noticeable, that this method does not make synthetic data itself more applicable for training purposes. In other words, we still have skewed data distribution in our dataset. Our approach is aimed at making synthetic data have more information about initial distribution, what can also benefit while training with accumulation, like in the mentioned case (Gerstgrasser et al., 2024). Another way of mitigating Model Collapse implies using so-called corrective functions (Gillman et al., 2024). This approach was widely tested on the human motion synthesis task and significantly less on image generation. Our research is aimed more at generating images and text.

## 3 Distillation for synthetic pipeline

Our approach involves employing a method analogous to self-distillation (Mobahi et al., 2020) to mitigate Model Collapse. Let $L_M$ be the loss function of some generative model $M$, training on dataset $D$. We denote the resulting trained by minimizing $L_M$ model as $M_0$. Additionally, we define $D_{M_0}$ as the dataset, sampled by $M_0$. Now, drawing parallels with self-distillation, we define the model $M_1$ as the student model and the model $M_0$ as the teacher model. Then $L_{(M_0,M_1)}$ is loss function between student and teacher outputs. We emphasize that loss can be computed not only between predictions of models but also, for instance, between outputs of particular layers.

We define the final loss as a weighed sum of the original loss function $L_M$ and the loss function between predictions of teacher and student: $L_{SD} = L_M + \lambda L_{(M_0,M_1)}$, where $\lambda$

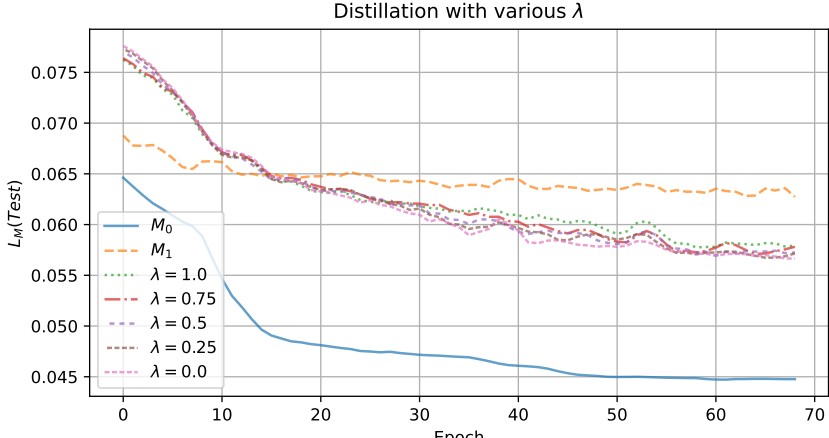

Figure 1: Comparison of distillation with various $\lambda$. An analysis of model performance relative to variations in the $\lambda$ coefficient reveals that setting $\lambda$ to zero does not significantly diminish the model's capabilities when compared to instances where $\lambda$ assumes nonzero values.

is the weight of distillation. We refer to this function as the loss function of synthetic distillation (SD). Then the model, trained by minimizing this loss, is denoted as $M_{distilled}$.

By minimizing $L_{SD}$, we simultaneously enable the student model to learn to solve the task in the conventional manner while also assimilating the skills acquired by the model trained on real data, which significantly enhances the model performance.

We note that we did not utilize data accumulation (Gerstgrasser et al., 2024); that is, the dataset used to train models subsequent to $M_0$ does not contain real-world data and maintains the same size.

## 4 Image generation

We now turn our attention to experiments involving image generating models. We will begin with a simple example of using the VAE (Kingma & Welling, 2013) architecture and subsequently explore more advance architecture DDPM (Ho et al., 2020).

### 4.1 VAE

We commence our examination of generative image models with the Variational Autoencoder (VAE). For our experiments, we selected MNIST (LeCun et al., 2010) dataset. We denote $L_{VAE} = L_{rec} + \lambda D_{KL}$, where $L_{rec}$ represents the reconstruction loss, $D_{KL}$ denotes KL-Divergence, $\lambda$ is distillation weight. Chosen architecture has encoder and decoder inside, each of which consist of two linear layers with ReLU nonlinearity between. Output is 32x32 pixels single-channel picture. Each of models, has approximately 887K parameters.

The training is conducted on train subset of MNIST size of 60000 examples, models are evaluated on test subset size of 10000 examples. For optimization AdamW optimizer (Loshchilov & Hutter, 2019) is used with learning rate 0.001. The training lasts for 70 epochs with a batch size of 256 pictures, equivalently 16450 optimization steps. Evaluation is performed after each epoch, in other words, on every 236th step.

After training $M_0$, we sample a dataset size of 60000, using the best weights in terms of the magnitude of loss function on test subset $L_M(Test)$, on which we conduct following experiments. This principle is also true for other architectures.

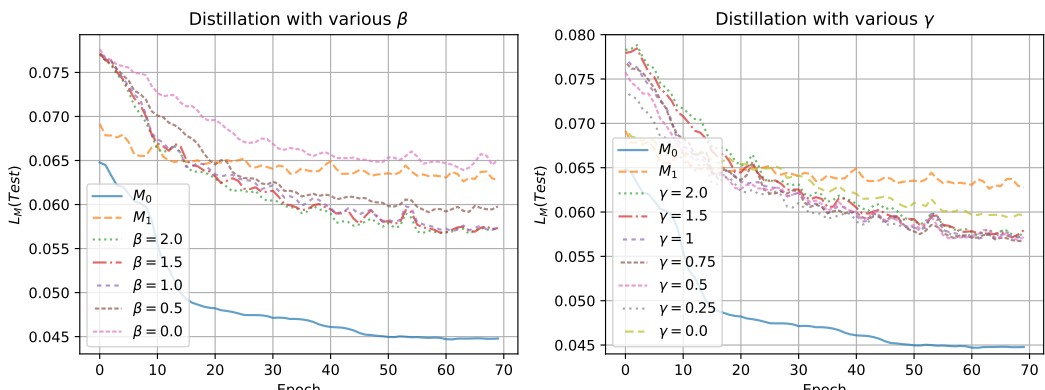

Figure 2: Comparison of distillation with various coefficients $\beta$ and $\gamma$. **Left**: comparison of different $\beta$. The less the $\beta$, the greater $L_M(Test)$, setting $\beta$ to zero leads to performance even worse, than our synthetic baseline $M_1$; **Right**: there is no clear conclusion about $L_M(Test)$, but $\gamma$ equals zero has the worst results, while other nonzero values show comparable results.

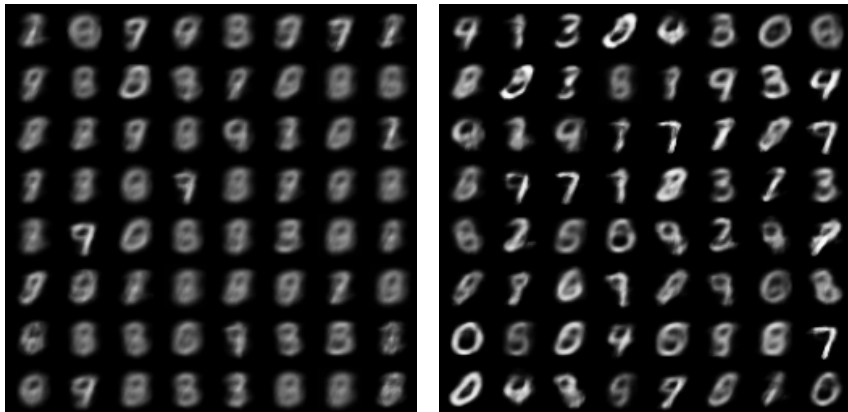

Figure 3: Comparison of samples generated by models. **Left**: samples generated by model $M_1$ (without using our method); **Right**: samples generated by model $M_{distilled}$.

Let $Y$ be ground-truth labels, $\hat{Y}_s$ be predictions of student ($M_1$), and $\hat{Y}_t$ be predictions of teacher ($M_0$). Moreover let $E_s, E_t, Var_s, Var_t$ be predictions of means and variances for reparametrization respectively.

Thus, $M_{distilled}$ is model, minimizing following loss:

$$L_{SD} = L_{VAE}(\hat{Y}_s, Y) + \lambda L_{VAE}(\hat{Y}_s, \hat{Y}_t) + \beta L_{MSE}(E_s, E_t) + \gamma L_{MSE}(Var_s, Var_t) \quad (1)$$

where, $\lambda$, $\beta$, $\gamma$ are some coefficients, $L_{VAE}$ is VAE loss function, $L_{MSE}$ is mean squared error (MSE).

The results of our approach are depicted in Figure 1, depending on $\lambda$, whilst other coefficients are fixed: $\beta = 1.0$, $\gamma = 1.0$. As a result, we have found out, that $L_{VAE}(\hat{Y}_s, \hat{Y}_t)$ term has little significance in terms of minimizing $L_M(Test)$. In general, we can see the greater $\lambda$, the worse the results.

Now we move on to $\beta$, other coefficients are fixed: $\lambda = 0.5$, $\gamma = 1.0$. As we can see in Figure 2 (**Left**), a decrease of $\beta$ leads to worse results. Setting it to zero shows performance even worse, than $M_1$. Increase of $\beta$ leads to better results to a certain extent.

Now consider coefficient $\gamma$. Comparison is in Figure 2 (**Right**), other coefficient are fixed: $\lambda = 0.5$, $\beta = 1.0$. No significant difference is noticed in range from 0.25 to 2.0. Nonetheless,

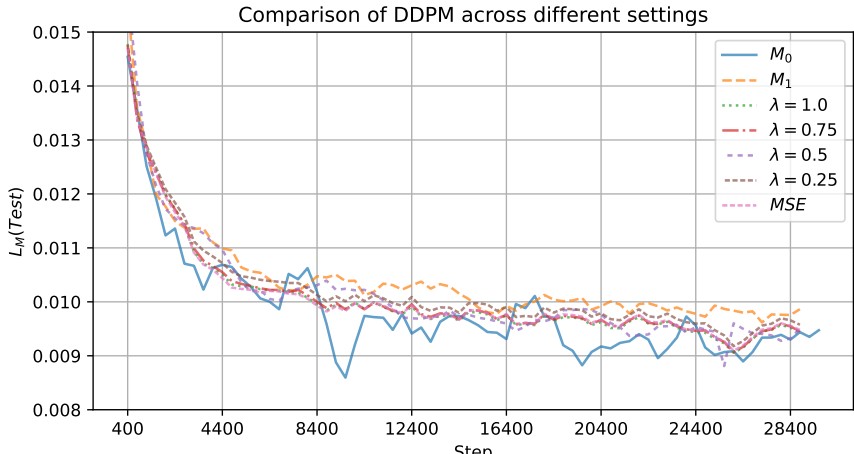

Figure 4: Results of our approach applied to DDPM architecture. Our method shows results comparable with the initial model $M_0$. In the legend, $MSE$ refers to the model that minimizes $L_{SD}$ while incorporating an additional MSE penalty between the encoder outputs.

setting to zero reduces model performance significantly, though the model still outperforms $M_1$.

Results for all models can be found in Table 1. Best model has following coefficient: $\lambda = 0.5$, $\beta = 1.0$, $\gamma = 1.0$. Meanwhile, setting $\lambda$ to zero leads to a loss of 0.12% regarding other coefficients being fixed in terms of $L_M(Test)$, $\beta$ to 12.91% and $\gamma$ to 4.83%. Last row corresponds to $M_1$, regarding this model, the performance has improved by 9.6%.

Besides quantitative results, samples of model $M_1$ and model $M_{distilled}$ are presented in Figure 3. As we can see, $M_{distilled}$ (**Right**) generates not only more various examples, but also more accurate.

Table 1: VAE results depending on coefficients values

| $\lambda$ | $\beta$ | $\gamma$ | $L_M(Test)$ |
|-----------|---------|----------|-------------|
| 1.0 | 1.0 | 1.0 | 0.05739 |
| 0.75 | 1.0 | 1.0 | 0.05692 |
| 0.5 | 1.0 | 1.0 | **0.05630** |
| 0.25 | 1.0 | 1.0 | 0.05639 |
| 0.0 | 1.0 | 1.0 | 0.05637 |
| 0.5 | 2.0 | 1.0 | 0.05639 |
| 0.5 | 1.5 | 1.0 | 0.05651 |
| 0.5 | 0.5 | 1.0 | 0.05851 |
| 0.5 | 0.0 | 1.0 | 0.06357 |
| 0.5 | 1.0 | 2.0 | 0.05694 |
| 0.5 | 1.0 | 1.5 | 0.05708 |
| 0.5 | 1.0 | 0.75 | 0.05651 |
| 0.5 | 1.0 | 0.5 | 0.05654 |
| 0.5 | 1.0 | 0.25 | 0.05677 |
| 0.5 | 1.0 | 0.0 | 0.05902 |
| 0.0 | 0.0 | 0.0 | 0.06228 |

## 4.2 DDPM

Next architecture is DDPM (Ho et al., 2020). The training was conducted on subset size of 15000 images, each size of 64x64 pixels, from CelebA dataset (Liu et al., 2015). We

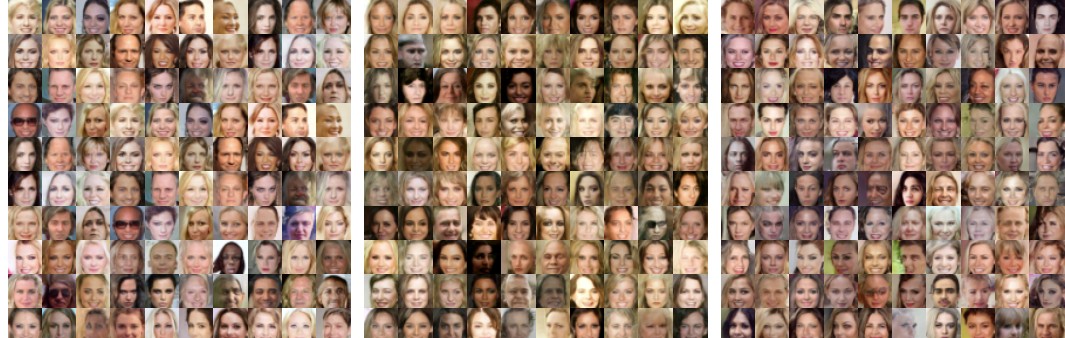

Figure 5: Results of our approach applied to DDPM architecture depicted in generated samples. **Left**: samples from $M_0$ are the most various, we can see different colors and even accessories on the face, but sometimes images are repeated. **Middle**: Samples from $M_1$. There are much less differences between generated images, people have predominantly blond hair and lighted faces. **Right**: Samples from $M_{distilled}$ with $\lambda$ set to 0.5. Our approach shows comparable results with high diversity in faces and hair.

take a batch of size 32 for training and 1 for evaluation. For optimization AdamW was used with learning rate 0.0001. For linear noise scheduler we use T equals 1000. Each model is trained for 30000 optimization steps, with evaluation on hold-out subset size of 5000 examples every 400 steps. Our model has default DDPM architecture of UNet (Ho et al., 2020), with attention intermediate layer and residual connections between Encoder and Decoder. Every model has 39M parameters.

As the DDPM loss function $L_{DDPM}$ we take SmoothL1Loss. Than our $L_{SD}$ has following form:

$$L_{SD} = L_{DDPM}(\hat{Y}_s, Y) + \lambda L_{DDPM}(\hat{Y}_s, \hat{Y}_t) \tag{2}$$

where, $Y$ are ground-truth labels, $\hat{Y}_s$ are predictions of $M_1$, $\hat{Y}_t$ are predictions of $M_0$, $\lambda$ is some coefficient.

The results of applying our method to the DDPM model are presented in Figure 4. As we can see, the distilled models consistently outperform $M_1$. We explore various values of $\lambda$, with the best performance observed at $\lambda = 0.5$. In addition to experimenting with $\lambda$, we also attempted to add to the Equation 2 MSE term between encoder outputs, scaled by 0.5 (denoted in Figure 4 as $MSE$), but this did not result in any improvement in performance. Furthermore, all models were used to sample 5000 images, which were then compared in terms of the Frechet Inception Distance (FID) (Heusel et al., 2018) with the evaluation subset. The complete results for all models are presented in Table 2. As shown, the model with $\lambda = 0.5$ yields the best results, first row with $\lambda = 0.0$ corresponds to the model $M_1$.

Table 2: DDPM experiments results.

| $\lambda$ | $L_M(Test)$ | FID |
|---|---|---|
| 0.0 | 93.49 $\times 10^{-5}$ | 59.94 |
| 1.0 | 89.53 $\times 10^{-5}$ | 49.35 |
| 0.75 | 89.82 $\times 10^{-5}$ | 54.04 |
| 0.5 | **88.09 $\times 10^{-5}$** | **48.81** |
| 0.25 | 90.98 $\times 10^{-5}$ | 52.24 |
| 0.5 $+MSE$ | 90.01 $\times 10^{-5}$ | 50.12 |
| $M_0$ | 83.32 $\times 10^{-5}$ | 43.77 |

Now we consider the qualitative enhancement, that provides our approach. In Figure 5 we can the the comparison between samples from $M_0$ (**Left**), $M_1$ (**Middle**) and $M_{distilled}$ with $\lambda$ set to 0.5. As we can see, our method helps to increase variety in generated samples, which means that the resulting distribution bears more resemblance to the initial one.

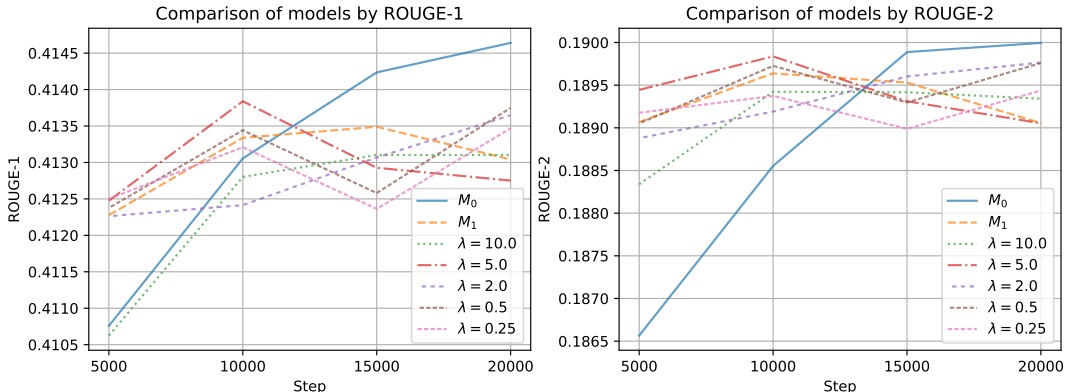

Figure 6: Results of our approach in terms of metrics ROUGE-1 and ROUGE-2. **Left**: Evaluation of ROUGE-1. As we can see, our models with $\lambda$ equals 5.0, 2.0, and 0.5 outperform $M_1$. **Right**: Evaluation of ROUGE-2. Our models mostly outperform $M_1$, in particular, model with $\lambda$ equals 5.0 shows significantly better results, comparable to $M_0$.

Overall, our approach substantially reduces the performance gap between $M_0$ and $M_1$, achieving a notable 68.8% improvement in terms of the FID metric and improves outputs diversity. These findings highlight the effectiveness of our method in enhancing model quality within the DDPM architecture.

## 5 Text summarization

Consider now our approach applying to causal language modelling task. In particular, we have chosen abstractive text summarization task. As a model we take small version of pretrained T5 (Raffel et al., 2023) and fine-tune it by minimizing $L_{CE}$ cross-entropy loss on newspaper dataset CNN/Daily Mail (See et al., 2017). Train subset includes 287K articles with corresponding highlights, validation subset has 13K examples. Training was conducted for 20000 steps with evaluation on every 5000th. Every article is cut to 1024 first tokens, every highlight to 128 first tokens, also during generation of synthetic data. For optimization we use a batch size of 32 articles, AdamW optimizer with learning rate 0.00002 and cosine scheduler with warmup for first 2500 steps and the number of waves equals 0.5. Overall, during training 655M tokens pass through the model. On every evaluation step we compute metrics ROUGE-1, ROUGE-2 and ROUGE-L (Lin, 2004). During generation we use greedy decoding.

Our $L_{SD}$ has the following form:

$$L_{SD} = L_{CE}(\hat{Y}_s, Y) + \lambda L_{CosDist}(\hat{Z}_s, \hat{Z}_t) \tag{3}$$

where, $Y$ are ground-truth labels, $\hat{Y}_s$ are student model ($M_1$) predictions, $\hat{Y}_t$ are teacher model ($M_0$) predictions, $\hat{Z}_s$, $\hat{Z}_t$ are outputs from the last layer of corresponding models, $L_{CosDist}$ is cosine distance and $\lambda$ is some coefficient.

Results of experiments in terms of metrics ROUGE-1 and ROUGE-2 can be found in Figure 6. As we can see, in case of ROUGE-1 (Figure 6 (**Left**)), where unigrams of sentences are compared, our approach shows better results than $M_1$, but distilled models still underperform $M_0$. Model with $\lambda$ equals 0.5 shows best results, while worst results has the model with $\lambda$ equals 10.0, which are even lower than with $\lambda$ being zero. In case of ROUGE-2 (Figure 6 (**Right**)), where bigrams are compared, we get some interesting results. We did not expect our approach enhance the results of ROUGE-2 more than ROUGE-1. Model with $\lambda$ equals 5.0 shows results, even comparable with $M_0$. We also can see similar behaviour in models with $\lambda$ equals 2.0 and 0.5. So, the best coefficients for enhancing ROUGE-1 results are between 0.5 and 5.0. Thus, the best coefficients for enhancing ROUGE-1 and ROUGE-2 are between 0.5 and 5.0.

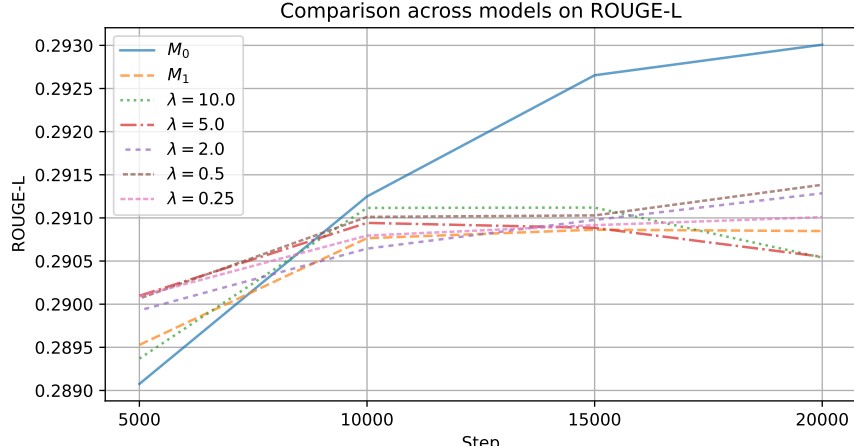

Figure 7: Results of models in terms of metric ROUGE-L. As we can see it is much more complicated for all models trained on synthetic data to close the gap. Nonetheless, our method improves the results of model $M_1$.

Results of testing models on metric ROUGE-L can be found in Figure 7. This metric is based on the longest common subsequence, what has certain implications. Our models do not perform as well as $M_0$, comparing with ROUGE-1 and ROUGE-2 cases, but still outperform $M_1$. Notably, the model with $\lambda$ set to 5.0 yields poorer results in terms of ROUGE-L metric. In contrast, $\lambda$ values of 0.5 and 2.0 demonstrate the best performance overall. In general, we think, that $\lambda$ should be between 0.5 and 2.0, for maximizing all the ROUGE metrics.

Table 3: Summarization experiments results.

| $\lambda$ | ROUGE-1, % | ROUGE-2, % | ROUGE-L, % | Mean perplexity |
|-----|-----------|-----------|-----------|----------------|
| 0.0 | 41.3489 | 18.9637 | 29.0863 | 30.6956 |
| 0.25 | 41.3469 | 18.9437 | 29.1009 | **32.0119** |
| 0.5 | 41.3749 | 18.9759 | **29.1384** | 31.8875 |
| 2.0 | 41.3647 | 18.9765 | 29.1285 | 31.7536 |
| 5.0 | **41.3839** | **18.9837** | 29.0942 | 31.3973 |
| 10.0 | 41.3102 | 18.9423 | 29.1119 | 31.4995 |

Besides ROUGE metric, we also evaluate perplexity per token for each model, with results presented in Figure 8. For this evaluation, we utilize the open HuggingFace library and the GPT-2 model (Radford et al., 2019) and use weights of our models, obtained after the last evaluation step. As illustrated, model $M_1$ truncates the tail of the tokens distribution and frequently generates tokens that the original model would not produce so often. In contrast, our model is less prone to this behavior, resulting in a broader distribution. The mean perplexity of each model, along with other evaluation metrics, is summarized in Table 3. Interestingly, the model with $\lambda = 0.25$ achieves the best perplexity performance, despite its relatively lower performance on the ROUGE framework. For clarity, we do not present histograms of perplexities of other models, as their distributions are approximately similar in shape.

The results of generating all models are shown in Table 4. As we can see, $M_1$ loses a lot in generation and falls into repeating the same phrase, while our model outputs a comparable summarization. We use version of distillation with $\lambda$ equals 0.25. Another example of generation can be found in Table 5. $M_1$ shows much better results, but our model $M_{distilled}$ generates summarization with more details.

In conclusion, our approach shows promising potential for improving performance in causal language modeling tasks. The evaluation using both ROUGE and Perplexity metrics reveals

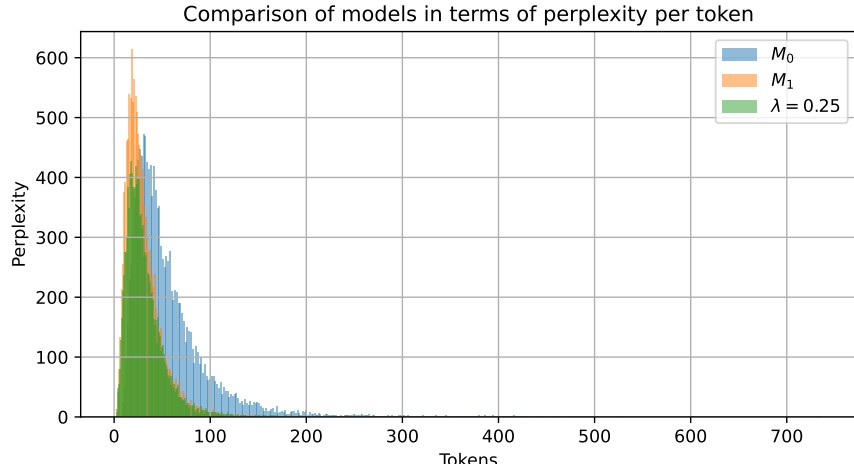

Figure 8: Perplexity per token in outputs of model $M_0$ (trained on real data), $M_1$ (trained on synthetic data) and our distilled version of $M_1$ with $\lambda = 0.25$. As we can see, our model demonstrates lower tendency to "narrowing" of distribution.

that, while ROUGE scores provide insight into the summary accuracy of the model, perplexity offers a deeper understanding of token-level generation quality. Notably, the model with $\lambda = 0.25$ achieves the best perplexity results, highlighting the effectiveness of our method, despite its relatively lower performance on ROUGE. These findings suggest that our model is less prone to undesirable token generation.

Table 4: Models outputs comparison.

| | |
|---|---|
| **Original highlight** | The average WNBA player makes \$72K; the average NBA player makes \$5 million. There are huge disparities between men's and women's sports, a former Olympic champ says. A recent survey says both men and women believe men are better at sports. |
| **$M_0$ output** | Women's sports have a long way to go before we get to true gender equality. The women are still young, with Title IX only a few decades old. The women are a story without a photo deep in the sports section. |
| **$M_1$ output** | Women's sports are the one area where they believe there are differences. The women? A story without a photo deep in the sports section. The women? A story without a photo deep in the sports section. |
| **$M_{\mathbf{distilled}}$ output** | The women's basketball team hasn't had a great season for years. The Princeton women's basketball team hasn't had a great season for years. The women's sports are the one area where they believe there are differences. The women's sports is still young, with Title IX only a few decades old. |

## 6  Discussion

In conclusion, our paper presents a novel approach to self-distillation, demonstrating that distillation between a model trained on real-world data (teacher) and one trained on synthetic data (student) can significantly enhance the performance of the last one. Our method, which is simple to implement, proves effective in mitigating Model Collapse in both image and text generation tasks. While the method has shown promising results, its impact on models utilizing data accumulation strategies (Gerstgrasser et al., 2024) remains an open

Table 5: Examples of summarization.

| | |
|---|---|
| **Original highlight** | Kenneth Golightly, 29, charged with aggravated robbery. Accused of holding up MedStar ambulance driver at knifepoint in Fort Worth, Texas. Video from inside the ambulance shows Golightly running a red light and doing speeds of more than 70mph before crashing through a fence. |
| **$M_0$ output** | Kenneth Golightly, 29, charged with aggravated robbery in connection to the armed hijacking of a MedStar ambulance in Fort Worth, Texas. Police say Golightly pulled a knife on the driver and forced him out of the ambulance. The vehicle eventually crashed through a metal fence. Golightly fled the ambulance on foot after the collision. Police say a Good Samaritan witnessed the crash and Golightly's getaway, and followed him until officers responded to the scene and placed him under arrest. |
| **$M_1$ output** | Kenneth Golightly, 29, was arrested last Wednesday for allegedly carjacking an ambulance and taking it on wild joyride. Police say Golightly pulled a knife on the driver and forced him out of the ambulance. He then took the vehicle on a high-speed ride that lasted several blocks. The vehicle eventually crashed through a metal fence. |
| **$M_{distilled}$ output** | Kenneth Golightly, 29, was arrested last Wednesday and charged with aggravated robbery in connection to the armed hijacking of a MedStar ambulance in Fort Worth, Texas. Police say Golightly pulled a knife on the driver and forced him out of the ambulance. He then took the vehicle on a high-speed ride that lasted several blocks. The vehicle eventually crashed through a metal fence. |

question. Although we hypothesize that our approach could offer improvements in such scenarios, further empirical investigation is needed. Overall, we have shown that our self-distillation method increases the utility of synthetic data for training, suggesting a practical solution for improving performance across a variety of generative tasks. Future work will focus on exploring these open questions and extending the applicability of approach to a broader range of models and tasks.

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
