# OpenReview forum: "Leveraging Knowledge Distillation to Mitigate Model Collapse"
_ICLR.cc/2025/Conference — ICLR 2025 Conference Withdrawn Submission_

### Official Review · Reviewer_dp5r · 2024-10-19

**Soundness:** 2
**Presentation:** 2
**Contribution:** 1
**Rating:** 3
**Confidence:** 3

**Summary:**

This paper provides a contribution to addressing the problem of model collapse using synthetic data. The proposed method leverages knowledge distillation to address this problem. Experiments on multiple image genearion models and text generation model are conducted to indicate the effectiveness of the proposed method.

**Strengths:**

1. The proposed method is easy to follow.
2. The structure of the paper is clear.

**Weaknesses:**

1. There is no formal definition about "Model Collapse" indicated in the paper, the author should describe it for both text model and image model in more details. Also I do not agree that the test set loss and ROUGE scores are a good metric for model collapse indication.
2. The adopted datasets for image generation are quite simple. The authors should use more complex datasets.
3. There is no theoretical/empirical analysis about the results and findings, the authors should think about adding these.
4. The proposed method is still worse than $M_0$ after training with longer steps for language models. The authors should analyze this more.
5. The authors have modified the template style, which could be problematic.

**Questions:**

Please refer to the Weakness section.

---

### Official Review · Reviewer_eNi5 · 2024-10-27

**Soundness:** 1
**Presentation:** 1
**Contribution:** 1
**Rating:** 1
**Confidence:** 5

**Summary:**

The paper explores the issue of model collapse. The authors propose a solution using knowledge distillation, Experiments across image generation tasks, including Variational Autoencoder (VAE) and Denoising Diffusion Probabilistic Model (DDPM), and text summarization show performance gains.

**Strengths:**

1. Applying KD to mitigate model collapse might be a possible solution.

**Weaknesses:**

This paper’s format does not follow the ICLR requirements. Additionally, the presentation is poor, lacking clear motivation and an introduction to the methodology. Many unnecessary figures that should have been placed in the appendix occupy a large portion of the main text, making the paper resemble an experimental report. Even so, it fails to reach ten pages. This submission appears extremely unprofessional.

**Questions:**

NA

---

### Official Review · Reviewer_q7eK · 2024-11-01

**Soundness:** 3
**Presentation:** 2
**Contribution:** 2
**Rating:** 3
**Confidence:** 3

**Summary:**

This paper presents a method that utilizes knowledge distillation to mitigate the adverse effects of synthetic data by enhancing the transfer of knowledge from high-performing teacher models to student models. Through extensive experiments, they improve the robustness and generalization capabilities of models trained on synthetic data.

**Strengths:**

-	This paper uses knowledge distillation as a solution to address model collapse.
-	This paper conducts experiments on image generation using VAE and DDPM, as well as text summarization using the T5 model.

**Weaknesses:**

-	The template of the article is not officially provided.
-	The models used in this paper are VAE and DDPM. Can more advanced models be used for image generation, and can the resolution of the generated images be improved? This can better prove the generalization of the proposed method.
-	Lack of comparison with existing approaches to mitigate model collapse.

**Questions:**

Is there no existing baseline for comparison?

---

### Official Review · Reviewer_hT3U · 2024-11-04

**Soundness:** 2
**Presentation:** 1
**Contribution:** 1
**Rating:** 1
**Confidence:** 4

**Summary:**

The authors use a knowledge distillation framework that utilizes a model trained on real data as a teacher to address the issue of model collapse in models trained on synthetic data.

**Strengths:**

- The proposed approach is highly intuitive. (but, too obvious.)

**Weaknesses:**

- Lack of novelty
  - They merely used the conventional knowledge distillation (KD) method. I think they should have compared various KD methods to identify a more suitable approach for resolving model collapse on the same task.
- Lack of references
- Design of experiments
  - Their experimentation is limited to adjusting the hyperparameters of the model.
  - For iterative cases (repeated cycles of data generation and model retraining, as you mentioned in Introduction), I think it is necessary to assess how severe the model collapse becomes and to what extent it can be resolved.

**Questions:**

- (Writing) Please modify sty file for ICLR 2025.
- (Writing) The line style looks unorganized

---

### Note · Authors · 2024-11-25

**Comment:**

Good day, first of all, thanks to all the reviewers for their feedback, they helped us look at our work through other people's eyes and see the flaws that it has. Unfortunately, we did not have time to make all the necessary amendments to the main content of the paper, and we do not see any point in showing another raw article, so we withdraw our article from the competition. Thank you for the great opportunity to see the shortcomings of our article, we will definitely take them into account and improve our work in the future.

**Withdrawal Confirmation:**

I have read and agree with the venue's withdrawal policy on behalf of myself and my co-authors.